# Overexpression of an Inositol Phosphorylceramide Glucuronosyltransferase Gene *IbIPUT1* Inhibits Na^+^ Uptake in Sweet Potato Roots

**DOI:** 10.3390/genes13071140

**Published:** 2022-06-24

**Authors:** Chong Liu, Mingku Zhu, Jian Sun

**Affiliations:** 1Institute of Integrative Plant Biology, School of Life Sciences, Jiangsu Normal University, Xuzhou 221116, China; mr6v587@163.com; 2Jiangsu Key Laboratory of Phylogenomics & Comparative Genomics, School of Life Sciences, Jiangsu Normal University, Xuzhou 221116, China

**Keywords:** Na^+^ homeostasis, transgenic root system, Na^+^ flux, Na^+^ fluorescent imaging, inositol phosphorylceramide glucuronosyltransferase

## Abstract

*IPUT1* is a glycosyltransferase capable of synthesizing the glycosyl inositol phosphorylceramide (GIPC) sphingolipid. The GIPC sphingolipid is a Na^+^ receptor on cell membranes which can sense extracellular Na^+^ concentrations, promote the increase in intracellular Ca^2+^ concentrations, and plays critical roles in maintaining intracellular Na^+^ balance. Therefore, the *IPUT1* gene plays an important role in the genetic improvement of crop salt tolerance. Herein, the *IbIPUT1* gene, which encodes an ortholog of *Arabidopsis AtIPUT1*, from sweet potato was cloned. *Agrobacterium rhizogenes*-mediated in vivo transgenic technology, non-invasive micro-measuring technology (NMT) and Na^+^ fluorescence imaging technology were then combined to quickly study the potential function of *IbIPUT1* in salt tolerance. The data showed that *IbIPUT1* was involved in the regulation of root cell Na^+^ balance, and the overexpression of *IbIPUT1* could not promote sweet potato root cell Na^+^ efflux under salt stress, but it could significantly inhibit the Na^+^ absorption of root cells, thereby reducing the accumulation of Na^+^ in root cells under salt stress. Additionally, Ca^2+^ efflux in transgenic root cells was slightly higher than that in control roots under salt stress. Collectively, an efficient transgenic method for gene function studies was established, and our results suggested that *IbIPUT1* acts as a candidate gene for the genetic enhancement of sweet potato salt tolerance.

## 1. Introduction

Among a series of abiotic stresses, salinity is one of the major adverse factors that hampers crop growth and development and reduces yield and quality worldwide [1,2]. The detrimental effects of salt on plants are due to the accumulation of sodium in the soil resulting in reduced water availability and the toxic effects of sodium and chloride ions on plants [3]. About nine billion hectares of land in the world are salinized, and 20% of arable land and 50% of irrigated land faces salinization, making soil salinization an increasingly serious global problem [4,5]. Therefore, cultivating salt-tolerant crops is an effective strategy to improve crop yield in salinized farmland. Sweet potato (*Ipomoea batatas* (L.) Lam) is the seventh most important food crop in the world, and although it can grow in variable climatic conditions and marginal lands, sweet potato productivity and quality are still often threatened by multiple abiotic stresses, including salinity stress [6]. Maintaining Na^+^ homeostasis at the cellular and the whole-plant levels is vital for the salt tolerance of plants [7]. Thus, the fast characterization of genes involved in Na^+^ homeostasis can promote the engineering of salt-tolerant sweet potato. For instance, our previous report revealed that the overexpression of phosphatidylserine synthase-encoding gene *IbPSS1* confers Na^+^ homeostasis and salt tolerance via improving Na^+^/H^+^ antiport activities in sweet potato [8].

Salt stress can trigger an increase in cytosolic Ca^2+^ concentrations, which will activate Ca^2+^-binding proteins to exclude Na^+^, and the salt-induced Ca^2+^ increase is thought to be relevant to the detection mechanism of salt stress [3,9,10]. Recently, the glycosyl inositol phosphorylceramide (GIPC) sphingolipids in the plasma membrane (PM) were identified as the first molecular components of a cell-surface Na^+^ receptor required for the NaCl-induced cytosolic Ca^2+^ increment and PM Na^+^/H^+^ antiporter activation in *Arabidopsis*. Additionally, GIPC sphingolipids mediate changes in cell membrane potential induced by salt stress, as well as salt-stress-adaptive physiological responses [11]. Inositol phosphorylceramide glucuronosyltransferase 1 (IPUT1) transfers glucuronic acid (GlcA) residue from GDP-GlcA to inositol phosphorylceramide (IPC) to form GIPCs. IPUT1 was divided into the glucuronosyltransferase subfamily 8 (GT8), and GT8 also includes functional proteins as glucosyltransferase (glycogenin), galactosyltransferase (LgtC), and galacturonosyltransferase (GAUT1) [12]. The *AtIPUT1* gene encodes inositol phosphorylceramide glucuronosyltransferase in the model plant *Arabidopsis*, which is required for the synthesis of GIPC sphingolipids [11]. Although glucuronosyltransferase is important for the enhancement of salinity tolerance in plants, their relevant functions in most plants remain unknown. Therefore, the continued exploration of the function of the *IPUT1* gene in other plants in response to salt stress and its role in Na^+^ and Ca^2+^ homeostasis is urgently needed.

In this study, a GT8-encoding gene *IbIPUT1* was cloned from sweet potato based on *Arabidopsis AtIPUT1* through a homologous cloning strategy. Then, an *A. rhizogenes*-mediated efficient transgenic root (TR) system, non-invasive micro-measuring technology (NMT) and Na^+^ fluorescence imaging were integrated to explore the potential function of the *IbIPUT1* gene in inhibiting Na^+^ uptake in sweet potato roots. Our data showed that although the overexpression of *IbIPUT1* could not improve sweet potato root cell Na^+^ efflux under salt stress, it could remarkably inhibit the Na^+^ absorption of root cells, thereby reducing the accumulation of Na^+^ in root cells under salt stress conditions. Thus, *IbIPUT1* has the potential to genetically improve the salt tolerance of sweet potato by reducing Na^+^ influx into root cells.

## 2. Materials and Methods

### 2.1. Plant Materials and Growth Conditions

The sweet potato variety utilized in this study was the purple sweet potato Xuzishu 8 (Zi 8) obtained from Xuzhou Sweetpotato Research Center (Xuzhou, China). The sweet potato seedlings were transplanted into plastic pots with mixed-nutrient soil (perlite: vermiculite: original soil 1:1:3) and placed in a clean greenhouse for cutting propagation. The temperature in the greenhouse ranged from 20 to 25 °C with a photoperiod of 16 h, and there was a photosynthetic photon flux density of 300 mol.m^−2^ s^−1^.

### 2.2. Cloning of IbIPUT1 Gene and Plasmid Construction

The total RNA from different tissues of Zi 8 was extracted using a TRIzol Reagent Kit (Shanghai, China) [13], the concentration of RNA was measured using a NanoDrop2000 instrument (Shanghai, China), and the integrity was checked via agarose gel electrophoresis. After purification, the HiScript II Q Select RT SuperMix (Vazyme, Nanjing, China) was used to synthesize the first-strand cDNA. The PrimeSTAR high-fidelity enzyme (Shanghai, China) was used to amplify the full-length coding sequence with the Zi 8 cDNA as a template. The amplification primers used were as follows: (1F: 5′-ATGGGATCTTTTAGAGGGAGCTTTGG-3′; 1R: 5′-TCATACACTCGTACTCCTCTGCGTG-3′), and the amplified product was purified using the Takara Mini gel recycle best gel recovery kit (Shanghai, China). The purified product was ligated into the PGEM-T vector for sequencing and comparison. Subsequently, the coding region of *IbIPUT1* and *DsRed* was inserted into a pCAMBIA0390 expression vector. Two constructs (*pCaMV35S::DsRed* and *pUBI.U4::IbIPUT1-CaMV35S::DsRed*) were used for *A. rhizogenes*-mediated transformation.

### 2.3. Multiple Sequence Alignment and Phylogenetic Tree Construction of Proteins

The full-length coding sequence of the *IbIPUT1* gene was submitted to the NCBI database under the accession number MN843750. Using *IbIPUT1* as the query condition, the protein sequence in *Arabidopsis thaliana* was searched in the *Arabidopsis* protein database using BlastP (version 2.11.0), and the e-value cutoff was 0.05. Then, the IPUT1 proteins from different plants were then subjected to multiple-sequence alignment using the ClustalW (version 2.0.10) default setting, and GeneDoc (version 2.0.10) was used to shade the conserved sequences. The GT8 domain and transmembrane domain were marked above the alignment. Finally, MEGA (version 7.0.14) software was employed to compare and analyze the IPUT1 protein sequence of *Arabidopsis*, corn, rice, tomato, grape, poplar and sweet potato, a phylogenetic tree was constructed using the neighbor-joining method, and the bootstrap value was set to 1000.

### 2.4. Genetic Transformation Mediated by A. rhizogenes K599

The two recombinant plasmids mentioned above were introduced into *A. rhizogenes* K599 using the freeze–thaw method. The Agrobacterium strains were inoculated into TY + 50 mg/L Kan liquid medium and placed on a shaker at 28 °C and 200 rpm for 12–14 h, and then, they were suspended in permeation buffer MES (10 mM MES-KOH, 10 mM MgCl_2_ and 100 μM acetosyringone; PH = 5.2) after centrifugation for enrichment. When the OD value reached 0.5, it was allowed to stand for 2–4 h. The sweet potato vines were cut with uniform growth (6–7 leaves) and stem length (about 15 cm long), and a medical syringe was used to inject 1 mL of the bacterial solution into the sweet potato cut stems, which were then transplanted into pots with mixed soil. After 3 weeks of cultivation in the greenhouse, the transgenic-positive plants were collected for further experiments. The transgenic root induction rate was calculated using the following method: (plants with transgenic root/total number of plants injected) × 100%.

### 2.5. Total RNA Extraction and qRT-PCR Analysis

The total RNA of adventitious roots (control) and transgenic roots that excited red fluorescence under the illumination of a fluorescent flashlight was extracted using the TRIzol method [13], and the first-strand cDNA was synthesized using reverse transcription, as described above. Primer 5.0 was used to design *IbIPUT1* primers for qRT-PCR analysis. qRT-PCR was carried out using the CFX96™ Real-Time System (Bio-Rad, Hercules, CA USA), as described in our previous report [14]. The *GAPDH* gene (genebank accession number: jx177362.1) was used as the internal reference gene, and the relative expression of the gene was calculated using the 2^−ΔΔCt^ method [15].

### 2.6. Visualization of Na^+^ in Root Cells of Sweet Potato

The Na^+^-specific fluorescent dye CoroNa-Green AM (Life Technologies, Shanghai, China) was used to observe the accumulation of Na^+^ in different areas of root tissues (adventitious roots and transgenic roots). After 24 h of NaCl treatment, the root tips of the plants (about 3 cm) were collected and transferred to a culture medium containing NaCl (200 mM), 20 μM CoroNa-Green AM and 0.02% polyuronic acid (Life Technologies, Shanghai, China) [8,16,17]. After incubation for 2 h, the roots were washed 3–4 times with distilled water and then observed and photographed with an Olympus BX63 (Tokyo, Japan) inverted fluorescence microscope. The semi-quantitative analysis of the fluorescence intensity of CoroNaTM Green was performed using Image Pro Plus 6.0 software.

### 2.7. Steady-State and Instantaneous Ion Flow Measurement

A non-invasive micro-measurement system (NMT-100-SIM-YG, Younger USA LLC, Amherst, MA, USA) was used to detect the ion flow, as described in our previous publications [8,16]. Sweet potato seedlings with TR (red fluorescence excited by fluorescent flashlight), AR (without red fluorescence) or *DsRED* transgenic roots (transgenic control) were selected. After being soaked in Hoagland solution (1/4 Hoagland solution) for 24 h, NaCl was added to a final concentration of 200 mM. The root tips (3 cm) were taken and transferred to the test solution (Na^+^: 0.1, 0.5 and 1.0 mM NaCl; 0.1, 0.5 and 1.0 mM CaCl_2_; pH = 5.7), rinsed 4–5 times and then placed in the test solution to balance for 30 min. This balancing step could reduce the influence of Na^+^ released from the outer body space of the root surface cortex on the experimental results [18]. Subsequently, they were transferred to a 50 mm transparent Petri dish containing 10 mL of fresh test solution and fixed. The root apex meristem, elongation area and mature zone were measured [16,17]. The root apex meristem (at 300, 400 and 500 μm from the root tip) is the root growth zone of plants and is the origin of all differentiated root tissues. The elongation area (at 1.5, 2.0 and 2.5 mm from the root tip) refers to the upper part of the root apex meristem, where cells generated from the apical meristem are gradually formed through growth and differentiation. In the mature zone (at 10, 12 and 15 mm from the root tip), the cytoplasm and nucleus of some cells gradually disappear, and these cells lose the middle transverse wall to form conduits. Each detection point in the three root zones was measured for 3–5 min, and the average value of ion currents in all of the test points represented the steady-state ion flow rate. The data collection and conversion in the inspection process adopted the software developed by Beijing Xuyue Science and Technology Co., Ltd(Beijing, China).

Additionally, TR, AR or *DsRED* transgenic root tips (3 cm) were collected and immersed in 10 mL of the fresh test solution described above for 30 min, and then, they were transferred to a 50 mm transparent culture containing 9 mL of fresh test solution. After they were fixed in the dish, they were measured for 3–5 min. Then, NaCl solution was added with a final concentration of 20 mM or 150 mM (10 mL) according to different needs, it was allowed to stand for 30–60 s [16,18,19], and we continued to carry out measurements for 20–30 min.

### 2.8. Statistical Analysis

The data were analyzed using one-way analysis of variance (ANOVA) with Student’s *t* test. “*”, “**” and “***” indicate significant differences at *p* < 0.05, *p* < 0.01 and *p* < 0.001, respectively.

## 3. Results

### 3.1. Molecular Characterization and Evolutionary Analysis IbIPUT1

The *Arabidopsis AtIPUT1* gene was reported to encode inositol phosphorylceramide glucuronosyltransferase, which is required for the synthesis of GIPC sphingolipids [11]. Thence, the *IPUT1* gene in crops may be a notable candidate gene for the genetic improvement of crop salt tolerance. Subsequently, a glucuronosyltransferase-encoding gene from sweet potato Xuzishu 8 (Hereafter abbreviated as Zi 8) was cloned using a homologous cloning strategy. Multiple-sequence alignment between this protein and homologous IPUT1 from *Arabidopsis*, corn, rice, tomato, grape and poplar showed that these proteins all contained a typical GT8 domain (glucuronosyltransferase) and six transmembrane (TM) domains (Figure 1A).

In order to better understand the relationship between this protein and other GT8 members, a phylogenetic tree was constructed using the complete amino acid sequence of *Arabidopsis* homologs. The results showed that this protein shared a closest evolutionary relationship with AtIPUT1 (AT5G18480) (Figure 1B). Therefore, this sweet potato gene was named *IbIPUT1*. The open reading frame (ORF) of *IbIPUT1* was found to be 1611 bp, encoding a 536-aa polypeptide, with a molecular weight of 60.4 kDa and a predicted isoelectric point (pI) of 8.89.

### 3.2. Overexpression of IbIPUT1 Reduces Na^+^ Accumulation in Sweet Potato Root Cells

In order to further clarify the role of the *IbIPUT1* gene in regulating the Na^+^ balance of sweet potato, the *pCambia0390-IbIPUT1-DsRed* overexpression vector was constructed, and the formation of transgenic roots was induced by *A. rhizogenes*-mediated in vivo transgenic technology established in our previous report [8]. After three weeks of the injection of *A. rhizogenes* into Zi 8 stems, it was found that most Zi 8 seedlings carried a large number of red fluorescent roots, suggesting that the vector was successfully transformed. Additionally, the induction rate of transgenic roots (TRs) in Zi 8 was about 75%, and the biomass of genetically modified roots was large enough (Figure 2A). In order to determine the expression level of the *IbIPUT1* gene in TR and adventitious roots (ARs, control) under different conditions, we performed salt treatment (200 mM NaCl for 24 h) and sampled them for qRT-PCR analysis. The results showed that the expression level of *IbIPUT1* in TRs was three times higher than that in ARs under both control and under salt stress conditions, proving that the *IbIPUT1* gene was successfully overexpressed. Additionally, we found that in both ARs and TRs, the *IbIPUT1* expression levels were not affected by salt stress compared to the controls (Figure 2B).

After 24 h of 200 mM NaCl treatment, the differences in Na^+^ accumulation amplitude in different root zones of ARs and TRs were detected using Na^+^-specific fluorescent dyes. The experimental results showed that the cells in the elongation zone of AR showed bright green fluorescence, suggesting that the accumulation of Na^+^ was relatively high. However, the green fluorescence in the cells of the same root zone of TRs was significantly weaker than that of ARs, indicating that the accumulation of Na^+^ in the root zone cells of TRs was lower than that in ARs (Figure 2C). Accordingly, fluorescence quantitative analysis data showed that the Na^+^ fluorescence intensity in TRs decreased by 42% compared with that in ARs (Figure 2D). This experimental evidence shows that the overexpression of *IbIPUT1* can reduce the accumulation of Na^+^ in root cells.

### 3.3. Overexpression of IbIPUT1 Does Not Affect Na^+^ Efflux under Salt Stress

In order to verify whether the overexpression of *IbIPUT1* promotes Na^+^/H^+^ antiport activity in sweet potato roots, we further measured the steady-state Na^+^ flux changes under salt stress using non-invasive micro-measuring technology (NMT). The results showed that the Na^+^ efflux of cells in different root zones (including the meristematic zone, elongation zone and mature zone) of TRs and ARs was significantly enhanced after salt stress, wherein the meristem zone had the highest amplitude of Na^+^ efflux, followed by the elongation zone, and the mature zone was the lowest (Figure 3A).

Unexpectedly, there was no significant difference in the Na^+^ efflux amplitude between *IbIPUT1* transgenic roots and adventitious roots (Figure 3A,B), indicating that *IbIPUT1* overexpression could not promote Na^+^/H^+^ antiport activity and Na^+^ efflux in the PM of root cells. The data suggested that the decrease in Na^+^ accumulation in *IbIPUT1*-overexpressing roots was not caused by the increase in Na^+^ efflux.

### 3.4. Overexpression of IbIPUT1 Inhibits Na^+^ Uptake in Sweet Potato Root Cells

According to a previous report, Na^+^ can bind to the negatively charged uronic acid residues of GIPCs outside the PM [11]. Therefore, the overexpression of *IbIPUT1* would increase the content of GIPCs, which would enhance Na^+^-binding sites on the outer side of the PM, thereby inhibiting the absorption of Na^+^ by root cells. To test this, the difference in Na^+^ uptake rate between *IbIPUT1*-overexpressing root cells and adventitious root cells under transient salt treatment was further determined. The maximum Na^+^ absorption rate of ARs after 20 mM NaCl treatment reached 45,000 pmol cm^−2^ s^−1^, while the maximum Na^+^ absorption rate of *IbIPUT1*-overexpressing root cells was only 11,000 pmol cm^−2^ s^−1^, which was largely weaker than that of ARs (the maximum Na^+^ absorption rate was reduced by more than 70%) (Figure 4A,B). In plants transformed with the *DsRed* (transgenic control), the Na^+^ uptake rates of TRs and ARs did not show significant differences (Figure 4C,D). The above experimental results showed that the overexpression of *IbIPUT1* might reduce the accumulation of Na^+^ in root cells by reducing the absorption of Na^+^ by sweet potato root cells instead of enhancing the activity of Na^+^ efflux.

### 3.5. The Effect of IbIPUT1 Overexpression on the NaCl-Induced Ca^2+^ Kinetics in Sweet Potato Root Cells

In *Arabidopsis*, salt-stress-induced PM Ca^2+^ channel activation and cytoplasmic Ca^2+^ concentration enhancement are dependent on *AtIPUT1*-mediated GIPC sphingolipid synthesis [11]. Therefore, further experiments were performed to determine the response of cell Ca^2+^ in the elongation zone of TRs and ARs to salt stress using NMT. As shown in Figure 5, both TRs and ARs showed weak Ca^2+^ efflux before salt stress, and there was no difference between the two. After salt stress (150 mM NaCl) treatment, both ARs and TRs showed the obvious enhancement of Ca^2+^ efflux, then Ca^2+^ efflux gradually decreased, and it returned to the control state after 20 min (Figure 5). We did not find enhanced Ca^2+^ influx in *IbIPUT1*-overexpressing roots. On the contrary, the Ca^2+^ efflux amplitudes in the elongation zone cells of *IbIPUT1*-overexpressing roots were slightly higher than those in adventitious roots under salt stress. Taken together, the results prove that *IbIPUT1* can be used as a candidate gene for the genetic improvement of salt tolerance in sweet potato and has important application potential.

## 4. Discussion

*A. rhizogenes*-mediated genetic transformation technology has been widely used in plant functional genomics research, especially in legumes [20,21]. Our previous report established an *A. rhizogenes*-mediated root transgenic system for sweet potato and showed that the overexpression of the phosphatidylserine-synthase-encoding gene *IbPSS1* improved salt-induced Na^+^/H^+^ antiport activities and enhanced PM Ca^2+^ channel sensitivity to salt stress in transgenic sweet potato roots [8]. Compared with *A. tumefaciens*-mediated genetic transformation technology, *A. rhizogenes*-mediated transformation technology cannot directly obtain regenerated plants but can rapidly induce the formation of transgenic hairy roots, which is useful for studying the function of root-specific expression genes. It is also very beneficial to quickly understand the cytological function of related genes [8,21]. For instance, an efficient and convenient in vivo transgenic hairy root induction system in woody plants was established before, which can quickly conduct functional studies of anthocyanin synthesis regulation gene and BiFC (bimolecular fluorescence complementation) experimental analysis [22].

The regulation of Na^+^ balance is very important to plant salt tolerance. An important goal of the genetic improvement of crop salt tolerance is to improve the regulation of Na^+^ homeostasis, including improving the Na^+^ efflux of root cell PM and the vacuolar compartmentalization ability of Na^+^, as well as inhibiting Na^+^ absorption and preventing the long-distance transport of Na^+^ [3,23]. CoroNa™ Green is a Na^+^-specific fluorescent probe which combines with Na^+^ and emit green fluorescence, and the fluorescence intensity is proportional to the Na^+^ content; thus, it is widely used in the research of plant Na^+^ balance regulation [4,24,25]. The measurement of Na^+^ flow rate based on non-invasive micro-measuring technology (NMT) is another common technical method to study the regulation mechanism of plant Na^+^ steady-state equilibrium [16,18,26]. NMT can accurately determine the rate of Na^+^ uptake and excretion by plant cells. Integrating the above in vivo transgenic system, root cell Na^+^ fluorescence imaging and NMT-based Na^+^ flow rate measurement will allow a short-term understanding of the role of target genes in regulating Na^+^ balance in sweet potato [8]. In this study, an in vivo transgenic system, qRT-PCR analysis, microscopic Na^+^ fluorescent imaging and NMT detection were used in combination. Based on our pipeline, characterization of the *IbIPUT1* gene potentially involved in Na^+^ homeostasis regulation was completed within 40 days. In summary, a convenient and fast pipeline was established to characterize the regulatory genes of Na^+^ homeostasis in sweet potato, which quickly provided important supporting information for the genetic improvement of sweet potato salt tolerance.

The discovery of plant Na^+^ receptor GIPCs is an important achievement in the research field of plant Na^+^ balance regulation and salt tolerance mechanisms. In *Arabidopsis*, *AtIPUT1* mediates the synthesis of GIPC sphingolipids. Under high salt stress, Na^+^ binds to the negatively charged uronic acid groups of GIPC sphingolipids outside the PM, activates PM Ca^2+^ channels and causes Ca^2+^ influx and increases in cytoplasmic Ca^2+^ concentration, thereby activating the SOS signal pathway and cytoplasm membrane Na^+^/H^+^ antiporter, which promotes the excretion of Na^+^ and the steady-state balance of Na^+^ in plants [11]. In this thesis, the *AtIPUT1* orthologous gene *IbIPUT1* from sweet potato was cloned, and the overexpression of *IbIPUT1* lowered the Na^+^ accumulation in salinized transgenic root cells through different cellular mechanisms. Surprisingly, NMT data showed that Na^+^/H^+^ antitransport activity in *IbIPUT1*-overexpressing root cells did not increase, but the initial Na^+^ absorption rate was greatly reduced. We speculated that the findings of this study may have been due to the following reasons: (1) the overexpression of *IbIPUT1* may increase the content of GIPCs on the membrane, and Na^+^ binds to the sphingolipid sphingolipids of GIPCs outside the PM more through charge. The clumps do not enter the cytoplasm, leading to decreased Na^+^ accumulation in root cells. We speculate that the overexpression of *IPUT1* increases the sphingolipid content of PM GIPCs and can act as a physical barrier preventing the passive entry of extracellular Na^+^ into plant root cells. (2). Although the sphingolipid content of GIPCs may be increased in the PM of *IbIPUT1*-overexpressing sweet potato roots, the abundance and distribution of PM Ca^2+^ channels coupled to GIPCs may not be accordingly altered; thus, the salt-induced Ca^2+^ influx and specific cytoplasmic Ca^2+^ s signaling and PM Na^+^/H^+^ antitransport activity did not increase with the overexpression of *IbIPUT1*. We tested the changes in Ca^2+^ flux in *IbIPUT1*-overexpressing root tissues under salt stress and found no significant increase in Ca^2+^ influx in transgenic root cells under salt stress, which confirmed our speculation. The effect of *IbIPUT1* overexpression on the Ca^2+^ transport activity of root cells under salt stress needs to be further elucidated via root cell protoplasts. Collectively, we proved that the overexpression of *IbIPUT1* affected the Na^+^ balance of sweet potato root cells, which opened a new range of applications for sweet potato functional genomics studies.

## 5. Conclusions

Collectively, the inositol phosphorylceramide glucuronosyltransferase gene *IbIPUT1* from sweet potato was isolated. The data showed that the overexpression of *IbIPUT1* could remarkably inhibit Na^+^ absorption and slightly enhance Ca^2+^ efflux in the root cells of transgenic sweet potato, thereby reducing the accumulation of Na^+^ in root cells under salt stress. Therefore, an efficient transgenic method for gene function studies was established, and our results suggested that *IbIPUT1* was involved in the regulation of root cell Na^+^ balance and *IbIPUT1* acts as a candidate gene for the genetic enhancement of sweet potato salt tolerance.

## Figures and Tables

**Figure 1 genes-13-01140-f001:**
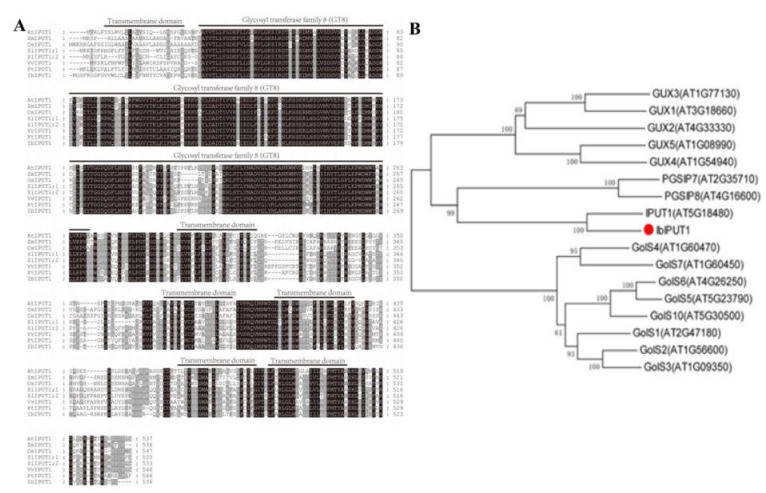
Molecular characterization of sweet potato IbIPUT1 protein. (**A**) Comparison of complete IPUT1 protein sequences in different species. The IPUT1 sequences included IbIPUT1 from sweet potato, AtIPUT1 from *Arabidopsis* (AT5G18480), ZmIPUT1 (GRMZM2G166903) from *Zea mays*, OsIPUT1 (LOC_Os02g41520) from *Oryza sativa,* SlIPUT1 (Solyc05g055040 and Solyc04g009920) from *Solanum lycopersicum*, VvIPUT1 (GSVIVG01023535001) from *Vitis vinifera* and PtIPUT1 (Potri 013G049100) from *Populus trichocarpa*. (**B**) Phylogenetic trees of IbIPUT1 and its homologs in *Arabidopsis*.

**Figure 2 genes-13-01140-f002:**
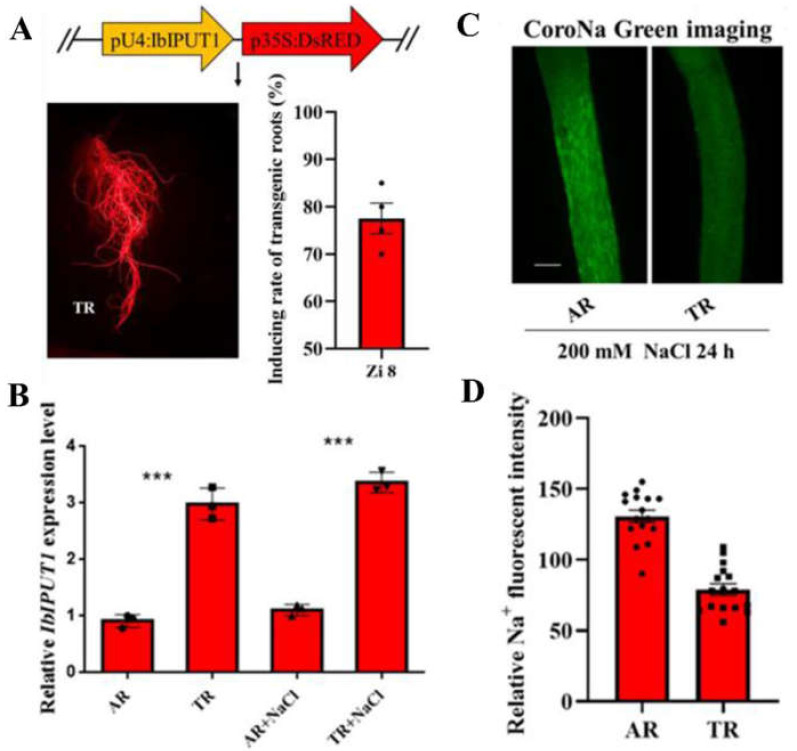
Overexpression of *IbIPUT1* influences cellular Na^+^ homeostasis in sweet potato roots. (**A**) Schematic of *IbIPUT1* expression cassette and representative images showing the *IbIPUT1*-transgenic roots (TRs) and non-transgenic adventitious roots (ARs) in the same seedling. Quantification of the transgenic rate of the vector in Zi 8 (*n* = 4). (**B**) The relative expression of *IbIPUT1* in transgenic roots (TRs) and adventitious roots (ARs). Internal reference: *GAPDH*. (**C**) Na^+^ accumulation in elongation root zone of ARs and TRs, as visualized through CoroNaTM Green (200 mM NaCl for 24 h). Bar = 0.3 mm. (**D**) Quantification of Na^+^ fluorescent intensity in (**C**) by using the Image-Pro Plus 6.0 software. Columns labeled with “***” indicate significant difference at *p* < 0.0001.

**Figure 3 genes-13-01140-f003:**
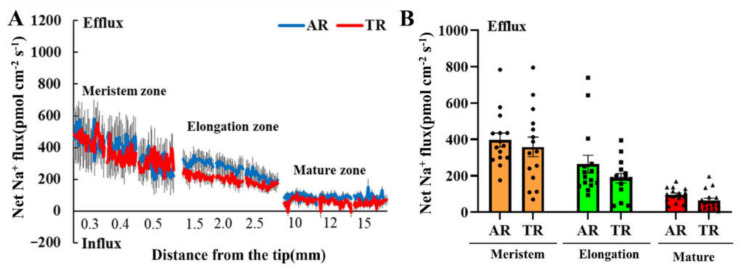
Effects of NaCl (200 mM) stress on steady-state Na^+^ flow in different regions of root tips of transgenic roots (TRs) and adventitious roots (ARs) of *IbIPUT1*. (**A**) Net Na^+^ flux measurement via NMT in adventitious roots (ARs) and *IbIPUT1*-transgenic roots (TRs). The steady-state Na^+^ flux was measured from the meristem (300, 400 and 500 µm from the tip), elongation (1.5, 2.0 and 2.5 mm from the tip) and mature (10, 12 and 15 mm from the tip) root zones, respectively, after 24 h of NaCl treatment (200 mM). Each measuring point is equivalent to the mean of at least 20 roots collected from 10 individual transgenic positive seedlings. (**B**) The histogram shows the average rate of net Na^+^ flux in the meristematic zone, elongation zone and mature zone. The error bar represents the standard error of the average value.

**Figure 4 genes-13-01140-f004:**
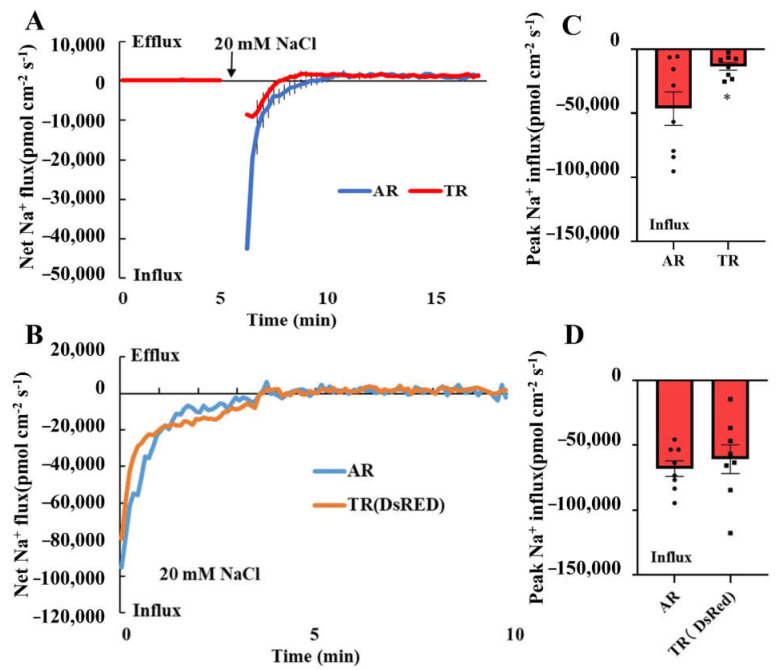
Overexpression of *IbIPUT1* inhibits Na^+^ uptake in sweet potato roots. (**A**) Transient Na^+^ influx upon treatment with 20 mM NaCl was measured from the elongation root zone (2 mm from the tip) of adventitious roots (ARs) and *IbIPUT1*-transgenic roots (TRs) through NMT. *n* = 8 from four seedlings. (**B**) Peak Na^+^ influx rate in (**A**). Columns labeled with “*” indicate significant difference between ARs and TRs at *p* < 0.05. (**C**) Transient Na^+^ flux upon 20 mM NaCl was measured via NMT from the elongation root zone (2 mm from the tip) of ARs and *DsRed*-transgenic roots (*n* = 7). (**D**) Peak Na^+^ influx rate in (**C**).

**Figure 5 genes-13-01140-f005:**
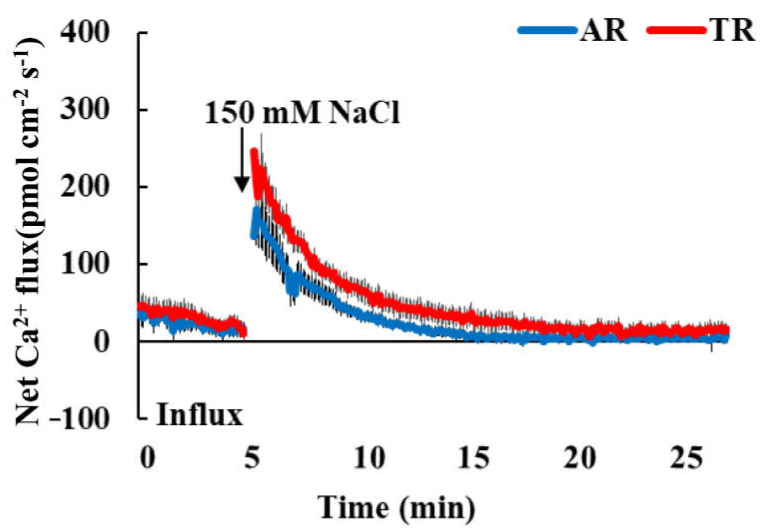
The effect of *IbIPUT1* overexpression on the NaCl-induced Ca^2+^ kinetics in the elongation zone of sweet potato. Transient kinetics of Ca^2+^ flux upon treatment with 150 mM NaCl was measured from the elongation root zone (1 mm from the tip) of TRs and ARs through NMT. *n* = 8.

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
