# Peer review of "Overexpression of an Inositol Phosphorylceramide Glucuronosyltransferase Gene IbIPUT1 Inhibits Na+ Uptake in Sweet Potato Roots"

_genes, 2022, doi:10.3390/genes13071140_

Round 1

Reviewer 1 Report

Authors cloned and performed the expression analysis of the IbIPUT1 gene in sweet potato by transient gene expression system. They successfully showed that IbIPUT1 was responsible for the Na+ absorption from the root cells. I have minor concerns about the manuscript.

1. Line 90, revise the sentence. Did you use A. tumefaciens or not?

2. How did you determine the stress duration. Is it enough to apply salt stress for just 24h?

3. Did you use the whole sequence or just the conserved GT8 domain sequence for alignment?

Author Response

Response to Reviewer 1

Authors cloned and performed the expression analysis of the IbIPUT1 gene in sweet potato by transient gene expression system. They successfully showed that IbIPUT1 was responsible for the Na+ absorption from the root cells. I have minor concerns about the manuscript.

Q1: Line 90, revise the sentence. Did you use A. tumefaciens or not?

Respond:

Thanks for your careful reminder about the use of Agrobacterium strains. Indeed, Agrobacterium Rhizogenesis-mediated transformation was used in our study. Then we have checked and corrected all the related descriptions in our revised manuscript. Thanks.

Q2: How did you determine the stress duration. Is it enough to apply salt stress for just 24 h?

Respond:

Thanks very much for your valuable and professional comments and suggestions, which are very helpful for improving our manuscript. According to our research experience and our previous published article, 24 or 48 hours of 200 mM salt treatment are suitable time points to detect the accumulation level of Na+ in sweetpotato roots. Based on your comments, the relevant references have been cited in our revised manuscript.

Q3: Did you use the whole sequence or just the conserved GT8 domain sequence for alignment?

Respond:

We are very grateful for your valuable comments. The whole sequence of IPUT1 protein sequences in different species were used for the alignment. The related description has been added in our revised manuscript.

In summary, we agree with the reviewer's nice and valuable comments and suggestions, and would like to express our great appreciation. We hope that this revision would make the manuscript more acceptable and suitable for publication. Thanks a lot.

Reviewer 2 Report

Dear Authors,

The Liu et al. manuscript describes an excellent synthesis.  In their paper, the authors have tested the hypothesis on Overexpression of an inositol phosphorylceramide glucu- 2 ronosyltransferase gene IbIPUT1 inhibits Na+ uptake in sweet- potato roots. I believe the importance of this paper stems from the applicability of the approach to saline conditions, the major adverse factors hampering crop growth and development, and reduces yield and quality worldwide. The results suggested that IbIPUT1 acts as a 25 candidate gene for the genetic enhancement of sweet potato salt tolerance.

In detail, the authors begin with Molecular characterization and evolutionary analysis IbIPTU1, followed by Overexpression of IbIPUT1/Na+ accumulation in sweet potato root cells; and Overexpression of IbIPUT1/Na+ efflux under salt stress. The effect of IbIPUT1 overexpression on the NaCl-induced Ca2+ kinetics in sweet potato root cells was also seriously assessed.

The main strengths of this paper are that it addresses an interesting and timely question, finds a novel solution based on a carefully selected set of rules, and provides a clear answer. As such this article represents an excellent and elegant bioinformatics genome-wide study that will almost certainly influence our thinking about the effect of salinity. 

Best regards

AK

Author Response

Response to Reviewer 2

The Liu et al. manuscript describes an excellent synthesis. In their paper, the authors have tested the hypothesis on Overexpression of an inositol phosphorylceramide glucu- 2 ronosyltransferase gene IbIPUT1 inhibits Na+ uptake in sweet-potato roots. I believe the importance of this paper stems from the applicability of the approach to saline conditions, the major adverse factors hampering crop growth and development, and reduces yield and quality worldwide. The results suggested that IbIPUT1 acts as a candidate gene for the genetic enhancement of sweet potato salt tolerance.

In detail, the authors begin with Molecular characterization and evolutionary analysis IbIPTU1, followed by Overexpression of IbIPUT1/Na+ accumulation in sweet potato root cells; and Overexpression of IbIPUT1/Na+ efflux under salt stress. The effect of IbIPUT1 overexpression on the NaCl-induced Ca2+ kinetics in sweet potato root cells was also seriously assessed.

The main strengths of this paper are that it addresses an interesting and timely question, finds a novel solution based on a carefully selected set of rules, and provides a clear answer. As such this article represents an excellent and elegant bioinformatics genome-wide study that will almost certainly influence our thinking about the effect of salinity.

Respond:

Thanks very much for your positive and constructive comments and suggestions on our manuscript, which are very helpful for improving our manuscript, and your encouragement will help us greatly in our future research. At the same time, we have further revised and improved the manuscript based on the comments of the reviewers. Thanks again.

In summary, we agree with the reviewer's nice and valuable comments and suggestions, and would like to express our great appreciation. We hope that this revision would make the manuscript more acceptable and suitable for publication. Thanks a lot.

Reviewer 3 Report

The manuscript deals with genes related to salt uptake in sweet potatoes and it could be useful for further experiments and genetic breeding. However, I have some concerns about the manuscript in its present form:

The main hypothesis and goals could be improved. They are not clear. One important hypothesis appeared in results section but the authors should find a better place for that.

The quality of some figures has to be improved.

In general, more detailed information in M&M section is needed to describe what is meristem, elongation, for instance... also a better description of statistics. This will help reviewers and readers to verify the soundness of your experiments.

Some sentences in results section could be removed or moved to Introduction

Please see comments on the MS file

Author Response

Response to Reviewer 3

The manuscript deals with genes related to salt uptake in sweet potatoes and it could be useful for further experiments and genetic breeding. However, I have some concerns about the manuscript in its present form:

Q1: The main hypothesis and goals could be improved. They are not clear. One important hypothesis appeared in results section but the authors should find a better place for that.

Respond:

Thanks very much for your positive and constructive comments and suggestions on our manuscript, which are very helpful for improving our manuscript. Based on your suggestions, the main hypothesis and goals have been improved in our revised manuscript. And the hypothesis appeared in results section has been modified in our revised manuscript. We sincerely hope that our improvements will meet with your approval. Thanks a lot.

Q2: The quality of some figures has to be improved.

Respond:

Thanks very much for your comments on this point. Indeed, the quality of some figures could be further improved, such as the low resolutions. Then we have improved the related figures according to your suggestions. Thanks very much.

Q3: In general, more detailed information in M&M section is needed to describe what is meristem, elongation, for instance... also a better description of statistics. This will help reviewers and readers to verify the soundness of your experiments.

Respond:

Thanks very much for your professional and constructive comments and suggestions on our manuscript, we have provided the related descriptions about meristem zones, elongation zones, mature zones, and statistical analysis according to your suggestions in our revised manuscript.

Q4: Some sentences in results section could be removed or moved to Introduction

Respond:

Thanks very much for your constructive suggestions on this point, which all are very helpful for improving our manuscript. Indeed, it would be more appropriate if some sentences in results section were removed or moved to Introduction section. We have made corresponding modifications in our revised manuscript, thank you very much.

Q5: Please see comments on the MS file

Respond:

Thanks for your careful review and generous revisions, we have revised or corrected all the related positions in our revised manuscript based on your markup. Thank you again for your careful revisions, which have helped a lot in improving our manuscript.

In summary, we agree with the reviewer's nice and valuable comments and suggestions, and would like to express our great appreciation. We hope that this revision would make the manuscript more acceptable and suitable for publication.  Thanks a lot.